# Designing and interpreting 4D tumour spheroid experiments

Ryan J. Murphy [1], Alexander P. Browning [1], Gency Gunasingh[2], Nikolas K. Haass [2,3] & Matthew J. Simpson [1,3 ✉]

Tumour spheroid experiments are routinely used to study cancer progression and treatment. Various and inconsistent experimental designs are used, leading to challenges in interpretation and reproducibility. Using multiple experimental designs, live-dead cell staining, and real-time cell cycle imaging, we measure necrotic and proliferation-inhibited regions in over 1000 4D tumour spheroids (3D space plus cell cycle status). By intentionally varying the initial spheroid size and temporal sampling frequencies across multiple cell lines, we collect an abundance of measurements of internal spheroid structure. These data are difficult to compare and interpret. However, using an objective mathematical modelling framework and statistical identifiability analysis we quantitatively compare experimental designs and identify design choices that produce reliable biological insight. Measurements of internal spheroid structure provide the most insight, whereas varying initial spheroid size and temporal measurement frequency is less important. Our general framework applies to spheroids grown in different conditions and with different cell types.

---

[1] Mathematical Sciences, Queensland University of Technology, Brisbane, QLD, Australia. [2] The University of Queensland Diamantina Institute, The University of Queensland, Brisbane, QLD, Australia. [3] These authors contributed equally: Nikolas K. Haass, Matthew J. Simpson. ✉email: matthew.simpson@qut.edu.au

Tumour spheroid experiments are an important in vitro tool routinely used since the 1970s to understand avascular tumour growth, cancer progression, develop cancer treatments, and reduce animal experimentation[1–10]. However, a vast range of experimental designs are employed, leading to inconsistencies in: (i) the times when measurements are taken; (ii) experimental durations, ranging from a few days to over a month[11–16]; (iii) the initial number of cells used to form spheroids[11–18], commonly between 300 to 20,000 cells[16,17]; and (iv) the type of experimental measurements that are taken[11–18]. This variability in experimental protocols makes comparing different studies very difficult, and introduces challenges in both interpretation and reproducibility of these experiments.

Mathematical modelling provides a powerful tool to provide such interpretation through model calibration and mechanism deduction. Simple mathematical models calibrated to outer radius measurements, such as Gompertzian growth models, have been used for decades to predict the growth of tumours[19,20]. However, these simple mathematical models do not provide information about the internal spheroid structure over time. In response, many mathematical models of varying complexity have been developed to explore the internal structure of spheroids[21–42]. Here, we revisit the seminal Greenspan mathematical model for avascular tumour spheroid growth[21] and, to the best of our knowledge, quantitatively directly connect it to data for the first time. Greenspan's mathematical model was the first to describe the three phases of avascular tumour spheroid growth: in phase (i) cells throughout the spheroid can proliferate; in phase (ii) cells near the periphery proliferate while a central region of living cells cannot proliferate, referred to as the inhibited region; and in phase (iii) there is an outer region of proliferative cells, an intermediate region of living inhibited cells, and a central necrotic region composed of dead cells and cellular material in various stages of disintegration and dissolution (Fig. 1a–d, Methods: Mathematical model). These various regions of cellular behaviour are thought to arise as a result of nutrient availability, such as oxygen, that is driven by diffusion and uptake. Using Greenspan's mathematical model has great advantages. All parameters are physically interpretable and biologically relevant, as opposed to more complicated mathematical models that may have parameters that cannot be interpreted physically and cannot be identified with the data in this study[27]. In addition, without using mathematical modelling and uncertainty quantification such that we employ here it is not obvious how to quantify the value of different experimental designs, and therefore impossible to interpret and determine the uncertainty of biologically relevant parameters.

In this study, we systematically explore a range of experimental designs and measurements. The first and simplest measurements we obtain are of the outer radius of the spheroid. Next, using live-dead cell staining we obtain measurements of the necrotic region. Measurements of the inhibited region are harder to obtain using traditional techniques. We use fluorescent ubiquitination-based cell cycle indicator (FUCCI) transduced cell lines[43–48]. The nuclei of these cells fluoresce red when cells are in the gap 1 (G1) phase of the cell cycle and green when cells are in the synthesis (S), gap 2 (G2) and mitotic (M) phases of the cell cycle (Fig. 1e). For clarity, we choose to show cells in the gap 1 (G1) phase in magenta instead of red. These data are collected for human melanoma cell lines established from primary (WM793b) and metastatic cancer sites (WM983b, WM164)[44,49–52], with endogenously low (WM793b) and high (WM983b, WM164) microphthalmia-associated transcription factor, which is a master regulator of melanocyte biology[53]. Analysing these data provides real-time visualisation of the cell cycle throughout tumour spheroids and powerfully reveals the time evolution of the

inhibited region (Fig. 1a–d). This additional dimension of information that we capture in our experiments, namely the cell cycle status, which can be thought of as a measure of time since a freely cycling cell enters the cell cycle, together with the three-spatial dimensions of the tumour spheroid give rise to the term 4D tumour spheroid experiments. Given an abundance of measurements of the outer radius, inhibited radius, and necrotic radius for tumour spheroids across multiple initial spheroid sizes, time points, and cell lines, we calculate maximum likelihood estimates (MLE) and form approximate 95% confidence intervals for the parameters of the Greenspan model. This allows us to quantitatively elucidate how modifying experimental designs can extract more information from experiments. Furthermore, this approach identifies the experimental design choices that are important and lead to reliable biological insight.

## Results

The results in this main document are for spheroids formed with the WM793b human melanoma cell line[44,50–52]. Additional results in Supplementary Discussion F and G show results for two other cell lines.

**Outer radius measurements are not sufficient to predict inhibited and necrotic radii.** Tumour outer radius measurements are simple to obtain and have been used for decades to quantify tumour growth[19,20]. Modern technology enables these measurements to be obtained more frequently, easily, and accurately. For example, the IncuCyte S3 live-cell imaging system (Sartorius, Goettingen, Germany) enables automated image acquisition and processing to measure spheroids every minute throughout an experiment providing a large number of measurements with ease. However, it is unclear whether these measurements provide sufficient information to understand and probe the internal structure of tumour spheroids and accurately predict tumour growth. Furthermore, it is unclear when measurements should be taken and the frequency of measurement. Performing experiments with WM793b spheroids formed with 5000 cells per spheroid, a typical choice in many experiments[7,11,53,54], 24 spheroids are imaged every 6 h. We monitor the time evolution of the outer radius to determine when spheroid formation ends and growth begins, which we call day 0 and occurs 4 days after seeding (Supplementary Discussion C.1.1), and to decide when to terminate the experiment, which we choose to be day 20. These measurements, supplemented with additional outer radius measurements from spheroids harvested for confocal imaging (Supplementary Discussion C.2–C.3), provide an abundance of data. We now compare three experimental designs with increasing temporal resolution: (i) Resolution A, using measurements from days 1, 3, 8, 12, 17 (Figs. 1l and 2a); (ii) Resolution B, using measurements from days 1, 3, 6, 8, 10, 12, 14, 17, 19 (Figs. 1m and 2b); and, (iii) Resolution C, using daily measurements from day 0 to day 19 (Figs. 1n and 2c). Excluding the final day(s) of measurements from these temporal resolutions allows a predictive check to be performed. Note that all these temporal resolutions are low relative to the capability of the automated imaging system but are high relative to the number of measurements typically taken in standard experiments[11,12,14,44,55].

To understand the influence of the choice of temporal resolution we now qualitatively and quantitatively compare the results. Across the three temporal resolutions in Fig. 2d–f we observe excellent agreement between the full set of outer radius measurements, collected every 6 h, and the predicted outer radius from the Greenspan model simulated with the MLE (Methods: Mathematical model, Practical parameter identifiability analysis). However, it is clear that the prediction of the inhibited and necrotic radius is poor with Resolutions A and B (Fig. 2d, e).

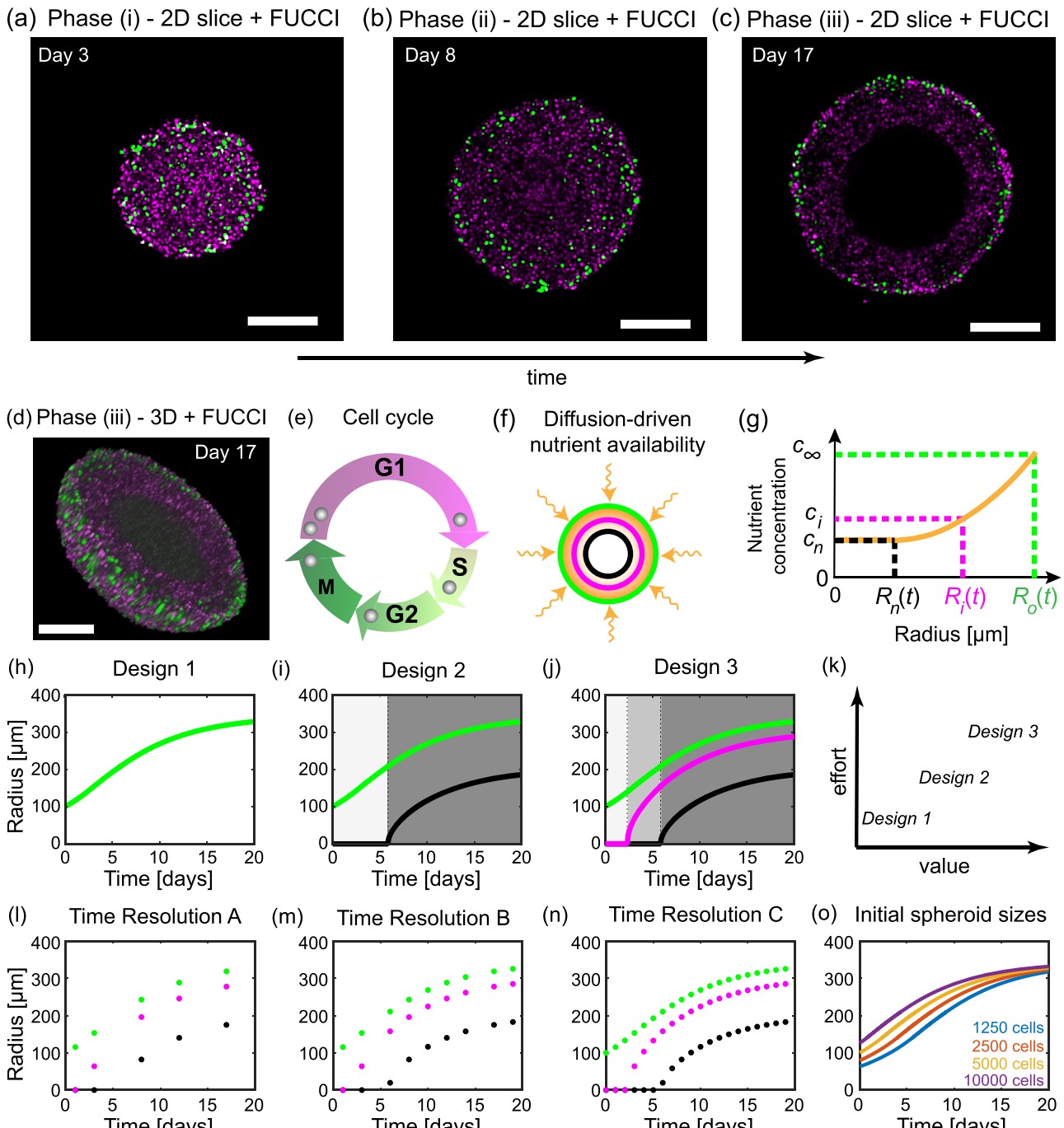

**Fig. 1 Tumour spheroid growth and the Greenspan mathematical model.** Tumour spheroids experience three phases of growth. **a–d** Confocal microscopy reveals different phases of tumour growth. Fluorescent ubiquitination-based cell cycle indicator (FUCCI) transduced cells allow visualisation of each cell's stage in the cell cycle. **a–c** 2D equatorial plane images of WM793b human melanoma tumour spheroids, formed with 5000 cells per spheroid, on days 3, 8, and 17 after formation. Scale bar 200 μm. **d** 3D representation of half of a WM793b human melanoma tumour spheroid on day 17 after formation, additional 3D representations are shown in supplementary discussion C.3.4. Scale bar 200 μm. **e** Cell cycle schematic coloured with respect to FUCCI signal. **f** Schematic for Greenspan mathematical model. Nutrient diffuses within the tumour spheroid and is consumed by living cells. **g** Snapshot of nutrient concentration, $c(r, t)$ for $0 < r < R_o(t)$, for a tumour spheroid in phase (iii) and where $R_o(t)$ is the tumour spheroids outer radius. External nutrient concentration is $c_\infty$. Inhibited radius, $R_i(t)$, and necrotic radius, $R_n(t)$, are defined as the radius where the nutrient concentration first reaches the thresholds $c_i$ and $c_n$, respectively. **h–j** Three experimental designs varying by measurement type. Design 1 considers only the outer radius (green). Design 2 considers the outer (green) and necrotic radius (black). Design 3 considers the outer (green), necrotic (black), and inhibited (magenta) radius. **k** Comparison of experimental designs with respect to their value and experimental effort required. **l–n** Three experimental designs that vary due to the time resolution at which measurements are taken. **o** Four experimental designs that vary the number of cells used to form each spheroid.

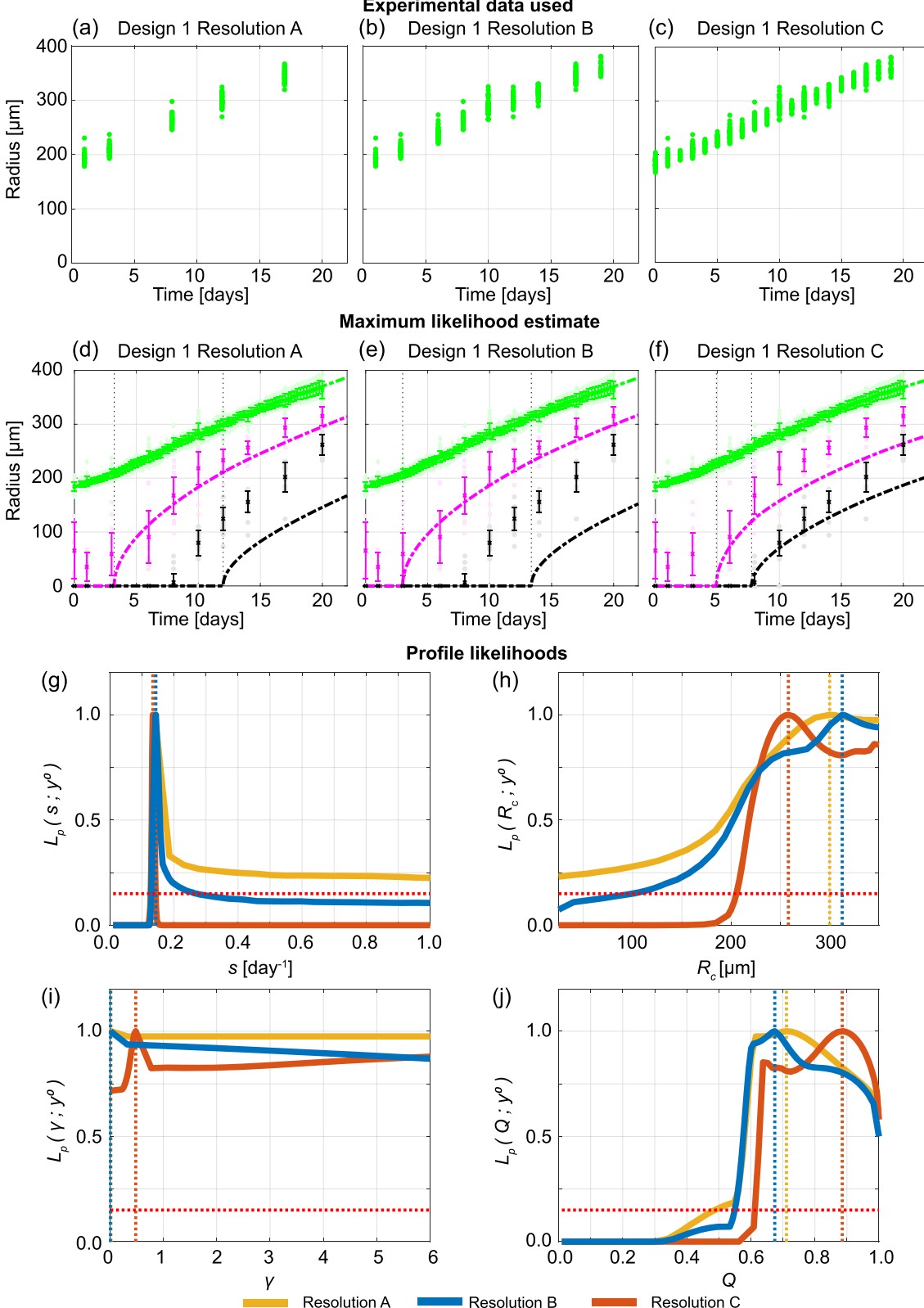

**Fig. 2 Increasing the temporal resolution when the outer radius is measured is not sufficient to predict necrotic and inhibited radii. a–c** Experimental data used in Design 1 with temporal resolutions A, B, and C. **d–f** Comparison of Greenspan model simulated with maximum likelihood estimate compared to full experimental data set, where error bars show standard deviation of the experimental data. Profile likelihoods for **g** $s$, **h** $R_c$, **i** $\gamma$, **j** $Q$. Yellow, blue, orange lines in **g–j** represent profile likelihoods from Design 1 with temporal resolutions A, B, and C, respectively, and the red-dashed line shows the approximate 95% confidence interval threshold. Results shown for WM793b spheroids formed with 5000 cells per spheroid.

With Resolution C, the prediction of the inhibited and necrotic radius appears to have improved (Fig. 2f) but we will show that it is misleading to suggest that increasing the temporal resolution is always beneficial. While MLE point estimates are insightful, it is unclear whether a similarly excellent match to the outer radius measurements could be obtained with different parameter values in the mathematical model. To answer this question we undertake a profile likelihood analysis of the five parameters that govern the behaviour of the mathematical model (Methods: Mathematical model):

1. $s$ [day$^{-1}$], the rate at which cell volume is produced by mitosis per unit volume of living cells (Fig. 2g),
2. $R_c$ [µm], the outer radius when the necrotic region first forms (Fig. 2h),
3. $\gamma = \lambda/s$ [-], the proportionality constant given by the rate at which cell volume is lost from the necrotic core, $\lambda$, divided by the rate at which cell volume is produced by mitosis per unit volume of living cells, $s$, (Fig. 2i),
4. $Q^2 = (c_\infty - c_i)/(c_\infty - c_n)$ [-], the ratio of the difference between the inhibited nutrient concentration threshold, $c_i$, and external nutrient threshold, $c_\infty$ to the difference between the necrotic nutrient concentration threshold, $c_n$, and external nutrient threshold, $c_\infty$ (Fig. 2j),
5. $R_o(0)$ [µm], the initial outer radius (Supplementary Discussion D.3).

Profile likelihoods are a powerful tool to visualise and analyse how many parameter values give a similar match to the experimental data in comparison to the MLE. Furthermore, we use profile likelihoods to compute approximate 95% confidence intervals for each parameter (Supplementary Discussion D.1). Narrow approximate 95% confidence intervals indicate parameters are identifiable and that few parameters give a similar match to the data as the MLE. In contrast, wide approximate 95% confidence intervals suggest that parameters are not identifiable, that many parameters give a similar match to the experimental data, and that additional information is required to confidently estimate the parameters.

The profile likelihoods for $s$ across all three temporal resolutions (Fig. 2g) lead to a peak that is close to $s = 0.14$ [day$^{-1}$]. These peaks correspond to the MLEs. While there is a wide 95% approximate confidence interval for $s$ with Resolution A, there are narrow approximate 95% confidence intervals for $s$ with Resolutions B and C. The profile likelihoods for the other parameters, $R_c$, $\gamma$, and $Q$, are wide and very similar using different temporal resolutions (Fig. 2h–j). For example, the profile likelihoods for $\gamma$ across all three temporal resolutions (Fig. 2i) are approximately flat and equal to one. These profile likelihoods for $R_c$, $\gamma$, and $Q$ suggest that increasing the temporal resolution provides little additional information. These results are consistent with additional results using synthetic data (Supplementary Discussion E). Additional results for different initial spheroid sizes (Supplementary Discussion D) and results for the WM983b cell line (Supplementary Discussion F.1) also clearly show that increasing the temporal resolution while using Design 1 may result in a worse prediction from the MLE for the time evolution of the internal structure. These results do not mean that the mathematical model is incorrect. Our interpretation of these results is that the experimental data are insufficient to correctly identify the parameters in the mathematical model. Overall, these results suggest that Design 1 (Fig. 1h) is not a reliable design to identify the true parameter values and cannot be used to determine details of the internal structure of tumour spheroids. This is important because this is the most standard measurement[5,14–16,19,55].

**Cell cycle and necrotic core measurements reveal time evolution of internal spheroid structure.** Given that measuring the outer radius of tumour spheroids alone (Design 1) is insufficient to determine details of the internal spheroid structure, we now examine which measurements are required to provide reliable estimates. The next simplest measurements to obtain are both the outer radius and necrotic core radius, which we refer to as Design 2 (Fig. 1i). However, Design 2 requires far more experimental effort since necrotic core measurements are more time-consuming involving harvesting, fixing, staining procedures, confocal microscopy or cryosectioning, and image processing. In addition, necrotic core measurements are end point measurements only, meaning that many spheroids are required to collect many data points. While intuitively we may anticipate that more experimental effort leads to more insight, it is impossible to quantify the value of this additional effort without a mathematical modelling and uncertainty quantification framework such that we employ here.

Using Design 2 for spheroids formed with 5000 cells per spheroid, we do not observe a necrotic core until approximately day 8 (Fig. 3a and Supplementary Discussion C.3.1). The Greenspan model simulated with the MLE obtained using temporal resolution A, since results obtained using temporal resolutions B and C are very similar (Supplementary Discussion D.2), excellently matches the growth of the outer radius, as before, and now captures the formation and growth of the necrotic core (Fig. 3c). Interestingly, the MLE suggests that the inhibited region is very small, so $R_i(t)$ is very close to $R_n(t)$. However, experimental measurements of the inhibited radius not only suggest that an inhibited region exists, but that it forms prior to the formation of the necrotic core (Supplementary Discussion C.3.1). Profile likelihoods for each parameter are relatively narrow, and because the profile for $Q$ is peaked and close to $Q = 1$, these profiles are consistent with either the absence of an inhibited region or a very small inhibited region (Fig. 3e–h). Therefore, these data do not identify the true parameter values since the calibrated mathematical model is inconsistent with the experimental observations that clearly show the formation of an inhibited region. This inconsistency does not mean that the mathematical model is incorrect. Our interpretation of this inconsistency is that this experimental data are insufficient to identify the parameters in the mathematical model.

We now explore Design 3, where we measure the outer, necrotic, and inhibited radius of multiple tumour spheroids (Figs. 1j and 3b). This design is considered third because measuring the inhibited radius is more difficult and requires substantial additional experimental effort. Using FUCCI-transduced cell lines in combination with optical clearing procedures and confocal microscopy powerfully reveals intratumoral spatiotemporal differences with respect to the cycle. This method also requires semi-automated image processing and expert guidance to minimise subjectivity and accurately identify the inhibited region boundary (Supplementary Discussion C.2)[54]. Simulating the Greenspan model with the MLE from Design 3 matches the evolution of the outer radius and captures the evolution of the necrotic and inhibited regions very accurately (Fig. 3d). Furthermore, the profile likelihoods for all parameters are well formed, with a single narrow peak, suggesting that Design 3 identifies the true parameter values (Fig. 3e–h). Comparing experimental Designs 1, 2, and 3, we observe that the profile likelihoods for $s$ are consistent across all designs (Fig. 3e) and the profile likelihoods for $R_c$ (Fig. 3f) are consistent for Designs 2 and 3. However, the profile likelihoods for $\gamma$ (Fig. 3g) and $Q$ (Fig. 3h) emphasise the power of measuring the inhibited radius and using Design 3 in comparison to Designs 1 and 2. These observations are consistent with additional results obtained using synthetic data (Supplementary Discussion E), different cell lines (Supplementary Discussion F), and initial

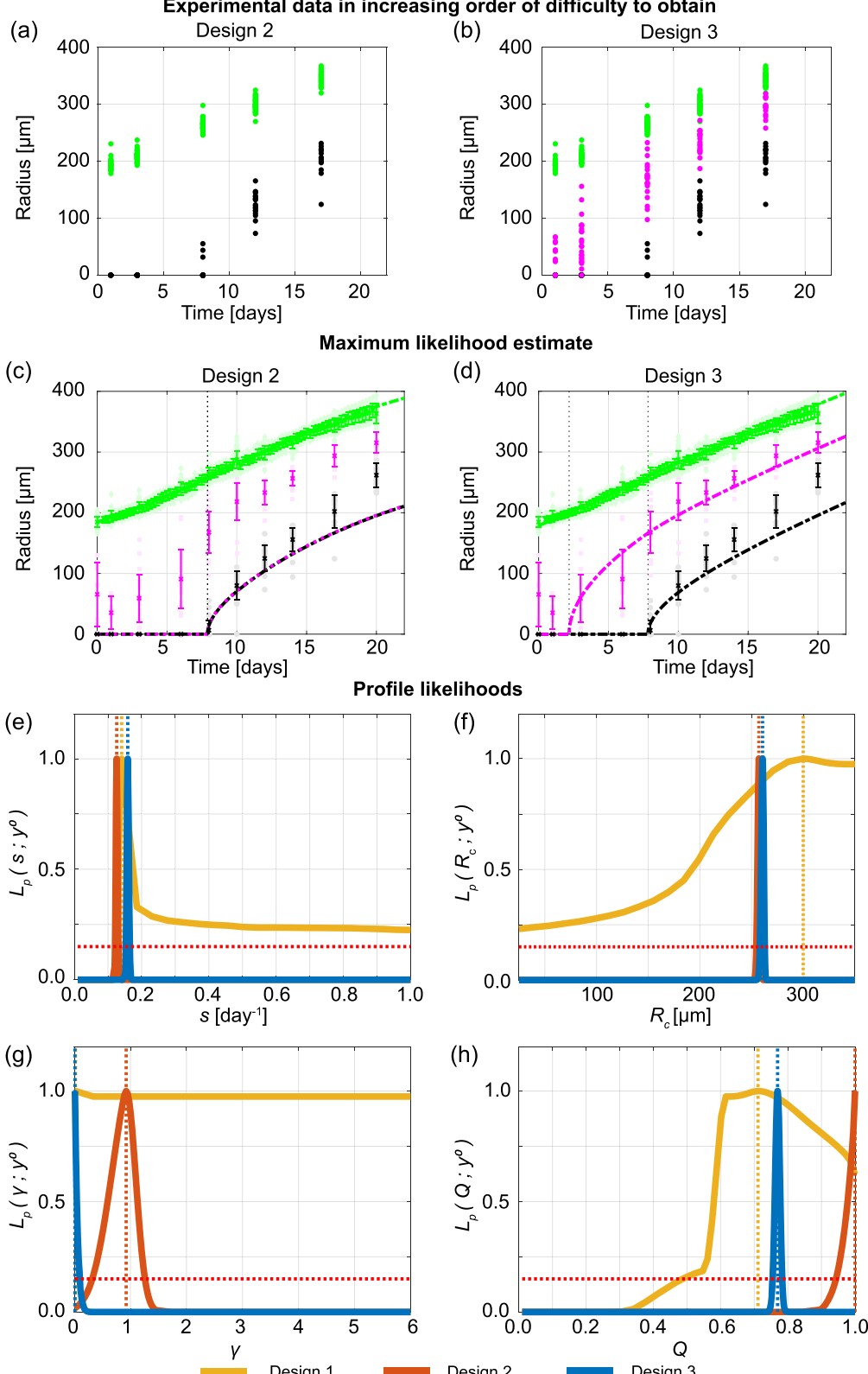

**Fig. 3 Measuring the necrotic and inhibited radius provides valuable information. a, b** Experimental data used in Designs 2 and 3 with temporal resolution A. **c, d** Comparison of Greenspan model simulated with maximum likelihood estimate compared to full experimental data set for Designs 2 and 3, where error bars show standard deviation. Profile likelihoods for **e** $s$, **f** $R_c$, **g** $\gamma$, **h** $Q$. Yellow, orange, blue lines in **e**–**h** represent profile likelihoods from Designs 1, 2, and 3, respectively, and the red-dashed line shows the approximate 95% confidence interval threshold. Results shown for WM793b spheroids formed with 5000 cells per spheroid.

spheroid sizes (Supplementary Discussion D.5). In Supplementary Discussion D.2, we also consider Design 3 with different temporal resolutions and experimental durations. As with Design 2, results obtained using Design 3 with temporal resolutions A, B, and C are very similar. Experiments performed for 4 or 10 days after spheroids form do not accurately predict late time behaviour. Designs that use days 10 to 20 or days 16 to 19 of measurements do not always accurately predict early time behaviour. Most insight is gained with Resolutions A, B, and C that cover the full experimental duration.

**Information gained using spheroids of different sizes is consistent**. In the literature tumour spheroids are initialised with a wide range of cell numbers, leading to inconsistent results that are difficult to meaningfully compare across different protocols[11–18]. Furthermore, it is unclear what the impact of this variability is when tumour spheroids are used to study fine-grained molecular-level interventions or potential drug designs. To quantitatively compare how information gained across experimental designs differs with respect to the initial number of cells in a spheroid we consider four initial spheroid sizes: 1250, 2500, 5000, 10,000 cells per spheroid (Fig. 1o). To proceed we use Design 3, and measure outer, necrotic, and inhibited radius, with time resolution A. Profile likelihoods for $R_o(0)$ show four distinct narrow peaks corresponding to each initial spheroid size as expected (Fig. 4a). Profile likelihoods for $s$, $R_c$, and $Q$ are consistent across the four initial spheroid sizes, allowing us to compare profile likelihoods on narrower intervals in Fig. 4b, c, e. The profile likelihoods for $\gamma$ (Fig. 4d) are more variable due to the differing number of measurements collected in phase (iii). These results suggest that the initial spheroid size does not play a significant role in determining information from experiments, provided sufficient measurements are obtained in phase (iii). To support these results, we show along the diagonal of Fig. 4f the solution of the mathematical model evaulated at the MLE associated with each initial spheroid size compared to the experimental measurements. Next, on the off-diagonals of Fig. 4f, we compare how the Greenspan model simulated with the MLE from one initial spheroid size predicts data from different initial spheroid sizes by only changing the initial radius. For example, in the top right of Fig. 4f we show that the Greenspan model simulated with the MLE obtained formed with 10,000 cells per spheroid agrees well with data from spheroids formed with 1250 cells per spheroid, when the initial radius is set to be the initial radius of the 1250 MLE. Results in Fig. 4f also show the inhibited and necrotic regions form earlier when considering spheroids formed with more cells, and results for spheroids formed with 10,000 cells per spheroid suggest that these spheroids form in phase (ii) rather than phase (i). These observations are consistent with additional results from synthetic data (Supplementary Discussion E) and the WM983b cell line (Supplementary Discussion F).

## Discussion

In this work, we present an objective theoretical framework to quantitatively compare tumour spheroid experiments across a range of experimental designs using the seminal Greenspan mathematical model and statistical profile likelihood analysis. By considering different temporal data resolutions, experiment durations, types of measurements, and initial spheroid sizes we identify the experimental design choices that lead to reliable biological insights and allow us to obtain reliable estimates with low uncertainty of parameters in Greenspan's model. This approach enables us to quantify the time evolution of the structure of growing spheroids and, since all parameters in Greenspan's model are physically interpretable and biologically

relevant, we also gain insights to the contribution of underlying biological mechanisms.

While it is common in spheroid experiments to measure the outer radius, it is less common to measure the necrotic core and not standard protocol to measure the cell cycle. However, we find that although Design 3, where we obtain outer, necrotic, and inhibited radius measurements, requires most experimental effort it is essential to determine the dynamics of tumour spheroid structure and growth. While we may have anticipated that more experimental effort leads to more insight, it is impossible to quantify the value of this additional effort without a mathematical modelling and uncertainty quantification framework such that we employ here. Alternative experimental designs may also provide useful insights but such approaches likely require more experimental effort and expense than the approach we use here of extending a typical experimental protocol to include necrotic core and cell cycle measurements. We note that should these alternative experimental designs be developed our framework will be relevant to their assessment.

We also show that temporal resolution and initial spheroid sizes are less important experimental design choices. Therefore, we recommend that for future studies, where tumour spheroid structure is important, that cell cycle data are essential and that some measurements using Design 3 are more valuable than many measurements using Designs 1 or 2. Furthermore, as information from tumour spheroids across varying initial spheroid sizes is relatively consistent, provided sufficient measurements in phase (iii) are obtained, we recommend that performing experiments with larger tumour spheroids can be beneficial to obtain useful information in a shorter experimental duration (Supplementary Discussion E.3). However, we also note that this may lead to large tumour spheroids that begin growth in phase (ii) rather than phase (i).

To perform this analysis we use Greenspan's seminal mathematical model, where all parameters have a relatively straight-forward biological interpretation. We find that Greenspan's model performs remarkably well across cell lines and initial spheroid sizes, and provides powerful insights into experimental design. Even though Greenspan's model is relatively simple, and may not capture all of the biological details of tumour spheroid growth, the fact that results for experimental data are consistent with those from synthetic data enhances our confidence that key biological features are captured in Greenspan's model (Supplementary Discussion E). Future modelling may wish to explore potential model misspecifications, for example, WM983b spheroids appear to reduce in size at very late time, suggesting a fourth phase in these in vitro experiments (Supplementary Discussion F); and, WM164 spheroids, possibly due to their lack of spherical symmetry[53], are more challenging to interpret as information gained using spheroids of different sizes is not consistent (Supplementary Discussion G).

The general framework presented in this work can be applied to other cell types, for example FUCCI-transduced lung, stomach and breast cancer cells[48,56–58], to extract more information from existing experimental data across experimental designs, and is suitable to be extended to consider tumour spheroids grown in different conditions and to more complex mathematical models. Given that cell cycle data is demonstrated to be informative in this study, we suggest that it may be beneficial for FUCCI technology to be further developed and more widely used, for example in avascular patient-derived organoids[59], and our framework be extended to these heterogeneous populations accordingly. Furthermore, the insights of this study provide a platform for future studies with spheroids that develop, test, and quantify the effectiveness of cancer treatments, possibly across different experimental designs. In such future studies cell cycle

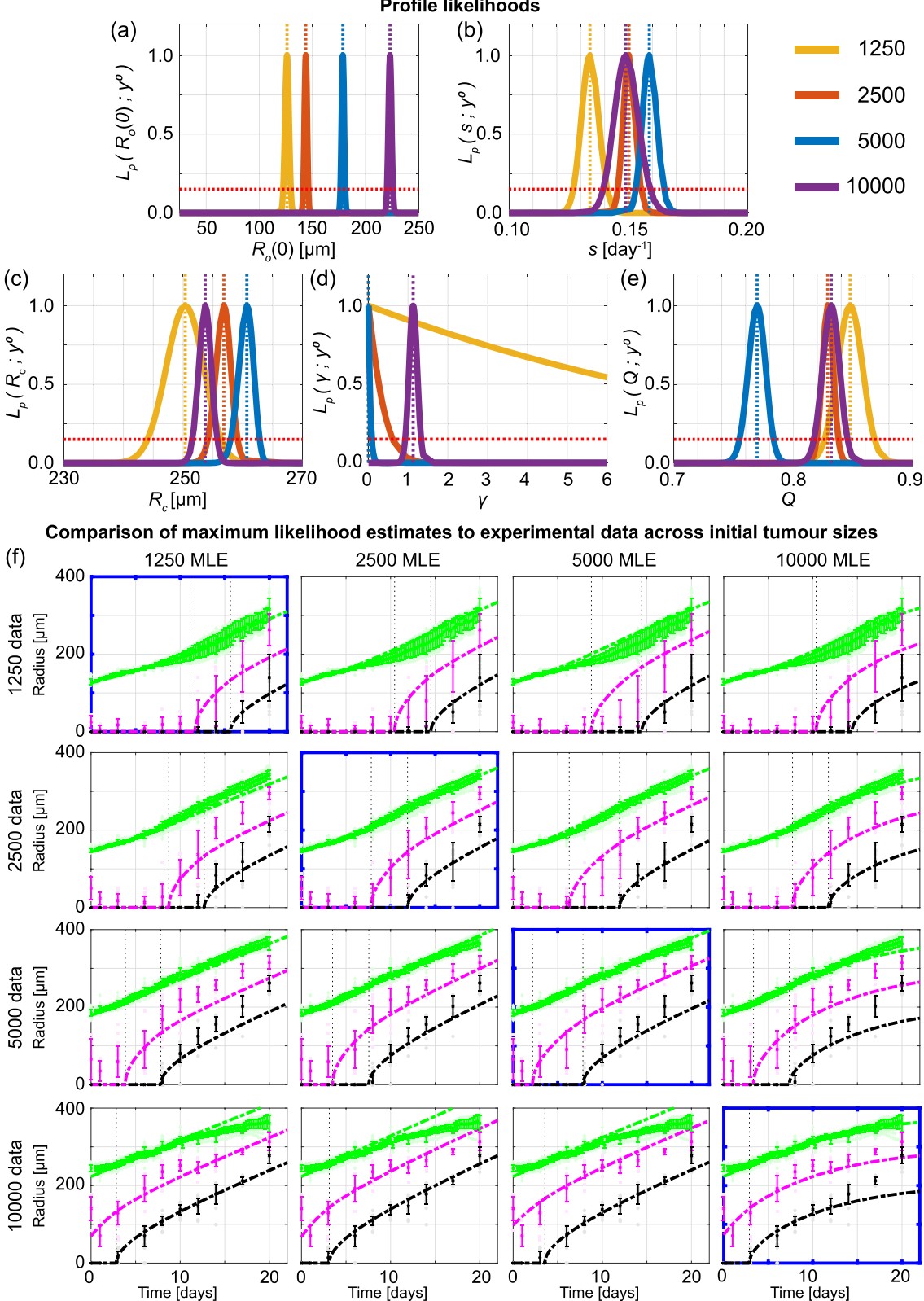

**Fig. 4 Information gained from experiments across different initial tumour spheroid sizes is mostly consistent.** Profile likelihoods for **a** $R_o(0)$, **b** $s$, **c** $R_c$, **d** $\gamma$, **e** $Q$. Yellow, orange, blue, and purple lines in **a**–**e** represent profile likelihoods from WM793b spheroids formed with 1250, 2500, 5000, 10,000 cells per spheroid, respectively, and the red-dashed line shows the approximate 95% confidence interval threshold. **f** Comparison of Greenspan model simulated with maximum likelihood estimates compared to full experimental data sets across initial tumour spheroid size, where error bars show standard deviation.

data will be informative since cytotoxic or cytostatic drugs may result in similar changes in the outer radius but due to different causes, that can be measured by cell death and cell cycle imaging (Haass laboratory personal communication). Therefore, understanding the time evolution of the outer radius, inhibited radius, and necrotic radius will be of great value in such future experiments, which is another advantage and contributing factor of developing a framework with Greenspan's model in this study.

## Methods

**Mathematical model.** Greenspan's mathematical model describes the three phases of avascular tumour spheroid growth[21]. Spherical symmetry is assumed at all times and maintained by adhesion and surface tension. Under these minimal assumptions, the only independent variables are time, $t$ [days], and radial position, $r$ [μm]. Tumour growth is governed by the evolution of the outer radius, $R_o(t)$ [μm], the inhibited radius, $R_i(t)$ [μm], and the necrotic radius, $R_n(t)$ [μm]. Nutrient diffuses within the spheroid with diffusivity $k$ [μm$^2$ day$^{-1}$] and is consumed by living cells at a constant rate per unit volume $\alpha$ [mol μm$^{-3}$ day$^{-1}$]. The external nutrient concentration is $c_\infty$ [mol μm$^{-3}$]. The nutrient concentration at a distance $r$ from the centre of the spheroid and time $t$, denoted $c(r,t)$ [mol μm$^{-3}$], is assumed to be at diffusive equilibrium. Therefore, at any instant in time we have $c(r,t) = c(r)$ due to fast diffusion of nutrient. However, as $R_o(t)$ is growing, nutrient diffusion occurs on a growing domain and we write $c(r) = c(r(t))$. The inhibited and necrotic regions form when the nutrient concentration at the centre of the spheroid reaches $c_i$ [mol μm$^{-3}$] and $c_n$ [mol μm$^{-3}$], respectively. For $c(r(t)) > c_n$ the rate at which cell volume is produced by mitosis per unit volume of living cells is $s$ [day$^{-1}$]. In the necrotic core cellular debris disintegrates into simpler chemical compounds that are freely permeable through cell membranes. The mass lost in the necrotic region is replaced by cells pushed inwards by forces of adhesion and surface tension. The necrotic core loses cell volume at a rate proportional to the necrotic core volume with proportionality constant $3\lambda$ [day$^{-1}$], where the three is included for mathematical convenience.

Conservation of mass is written in words as $A = B + C - D - E$, where $A$ is the total volume of living cells at any time, $t$; $B$ is the initial volume of living cells at time $t = 0$; $C$ is the total volume of cells produced in $t \geq 0$; $D$ is the total volume of necrotic debris at time $t$; $E$ is the total volume lost in the necrotic core in $t \geq 0$. Writing $A, B, C, D, E$ in their mathematical form gives the conservation of mass equation and also writing the nutrient diffusion equation gives,

$$R_o^2(t)\frac{dR_o(t)}{dt} = \frac{s}{3}\left[R_o^3(t) - R_i^3(t)\right] - \lambda R_n^3(t), \tag{1.1}$$

$$\frac{1}{r^2}\frac{\partial}{\partial r}\left(r^2\frac{\partial}{\partial r}c(r(t))\right) = \frac{\alpha}{k}\,\mathrm{H}\left(r - R_n(t)\right)\mathrm{H}\left(R_o(t) - r\right), \quad 0 \leq r \leq R_o(t) \tag{1.2}$$

where $R_i(t), R_n(t)$ are the radii implicitly defined by $c(R_i(t), t) = c_i$, and $c(R_n(t), t) = c_n$, respectively, if the nutrient concentration inside the spheroid is sufficiently small otherwise $R_i(t) = 0$ or $R_n(t) = 0$, and $\mathrm{H}(\cdot)$ is the Heaviside step function. There are eight unknowns: $\Theta = (s, \lambda, \alpha, k, c_\infty, c_i, c_n, R_o(0))$. Note this includes $R_o(0)$, which we treat as a parameter since we also need to estimate this quantity. Rescaling reduces the number of parameters to five: $\theta = (R_o(0), R_c, s, \gamma, Q)$. The new dimensionless parameters are: the outer radius when the necrotic region first forms defined as $R_c^2 = (6k/\alpha)(c_\infty - c_n)$; $Q^2 = (c_\infty - c_i)/(c_\infty - c_n)$; and $\gamma = s/\lambda$. Further details, and a formal demonstration that this model is equivalent to a model where nutrient determines the necrotic region and waste produced from live cells determines the inhibited region, are provided in Supplementary Discussion A.

**Practical parameter identifiability analysis.** To determine the maximum likelihood estimate (MLE) and approximate 95% confidence intervals for the parameters $\theta = (R_o(0), R_c, s, \gamma, Q)$ we use profile likelihood identifiability analysis[60–64]. We first choose simple parameter bounds and then compare the width of these simple parameter bounds to realised interval estimates for the parameters. Initial parameter bounds are chosen to be the same across all experimental designs analysed in this study. Outer radius data suggests we choose $0 < R_o(0) < 350$ [μm] and $0 < R_c < 250$ [μm]. Assuming a cell doubling time of at least 12 h and after performing initial simulations with a range of bounds, we set $0 < s < 1$[day$^{-1}$] (Supplementary Discussion B.1). Limited information exists for the parameter $\gamma$ so bounds were determined by performing initial simulations with a range of bounds before setting $0 < \gamma < 6$. By definition of $Q$ and experimental results that show the inhibited region forms before the necrotic core, we set $0 < Q \leq 1$. Note that the time evolution of $R_o(t)$ and $R_n(t)$ are the same for $Q = 1$ and $Q > 1$. The difference arises for $R_i(t)$, where it is equal to $R_n(t)$ for $Q = 1$ and equal to zero for $Q > 1$.

To determine the interval estimates for the parameters we treat the mathematical model as having two components. The first is the deterministic mathematical model governing the evolution of $R_o(t), R_n(t)$, and $R_i(t)$ and the second is a probabilistic observation model accounting for experimental variability and measurement error. Specifically, we assume that experimental measurements are noisy observations of the deterministic mathematical model[64,65]. For each of

the three measurement types $R_o(t), R_n(t)$, and $R_i(t)$ we assume that the observation error is independent and identically distributed and that the noise is additive and normally distributed with zero mean and variance $\sigma_o^2, \sigma_n^2$, and $\sigma_i^2$, respectively[65,66] (Supplementary Discussion B.3). We approximate $\sigma_o^2 \approx s_o^2, \sigma_n^2 \approx s_n^2$, and $\sigma_i^2 \approx s_i^2$ where $s_o^2, s_n^2$, and $s_i^2$ are pooled sample variances of the outer, necrotic, and inhibited radius measurements, respectively[67].

The likelihood function $p(y^o|\theta)$ is the likelihood of the observations $y^o$ given the parameter $\theta$. This corresponds to the probabilistic observation model evaluated at the observed data. The maximum likelihood estimate is $\hat{\theta} = \arg\max_\theta p(y^o|\theta)$. We present results in terms of the normalised likelihood function $L(\theta; y^o) = p(y^o|\theta)/\max_\theta p(y^o|\theta)$, which we consider a function of $\theta$ for fixed $y^o$. Profile likelihoods for each parameter are obtained by assuming the full parameter $\theta$ can be partitioned into a scalar interest parameter, $\psi$, and vector nuisance parameter, $\phi$, so that $\theta = (\psi, \phi)$. The profile likelihood for $\psi$ is then $L_p(\psi; y^o) = \max_\phi L(\psi, \phi; y^o)$. Approximate 95% confidence intervals are then calculated using a profile likelihood threshold value of 0.15 (Supplementary Discussion D.1)[62]. Prediction intervals are not shown since confidence intervals are narrow in many cases. Further details, including exact forms of the likelihood function and the use of log-likelihoods for calculations, and numerical methods are provided in Supplementary Discussion B.

## Experimental methods

*Cell culture.* The human melanoma cell lines WM793b, WM983b, and WM164 were provided by Prof. Meenhard Herlyn, The Wistar Institute, Philadelphia, PA[49]. All cell lines were previously transduced with fluorescent ubiquitination-based cell cycle indicator (FUCCI) constructs[7,44]. Cell lines were genotypically characterised[44,50–52], and authenticated by short tandem repeat fingerprinting (QIMR Berghofer Medical Research Institute, Herston, Australia). The cells were cultured in melanoma cell medium ("Tu4% medium"): 80% MCDB-153 medium (Sigma-Aldrich, M7403), 20% L-15 medium (Sigma-Aldrich, L1518), 4% fetal bovine serum (ThermoFisher Scientific, 25080-094), 5 mg mL$^{-1}$ insulin (Sigma-Aldrich, I0516), 1.68 mM CaCl$_2$ (Sigma-Aldrich, 5670); in an incubator (37 °C, 5% CO$_2$)[7]. Cell lines were checked routinely for mycoplasma and tested negative using the MycoAlert MycoPlasma Detection Kit (Lonza) and polymerase chain reaction[68].

*Spheroid generation, culture, and experiments.* Spheroids were generated in 96-well cell culture flat-bottomed plates (3599, Corning), with four different seeding densities (1250, 2500, 5000, 10,000 total cells/well), using 50 μL total/well non-adherent 1.5% agarose to promote formation of a single spheroid per well[54]. For all spheroid experiments, after a formation phase of 4, 3 and 2 days for WM793b, WM164 and WM983b, respectively (Supplementary Discussion C.1.1), and then every 3–4 days for the duration of the experiment, 50% of the medium in each well was replaced with fresh medium (200 μL total/well). Incubation and culture conditions are detailed in "Cell culture."

To estimate the outer radius, one plate for each cell line, containing 24 spheroids for each initial spheroid size, was placed inside the IncuCyte S3 live-cell imaging system (Sartorius, Goettingen, Germany) incubator (37 °C, 5%CO$_2$) immediately after seeding the plates. IncuCyte S3 settings were chosen to image every 6 h for the duration of the experiment with the 4 × objective. To estimate the radius of the inhibited and necrotic region and additional outer radius measurements, spheroids maintained in the incubator were harvested, fixed with 4% paraformaldehyde (PFA), and stored in phosphate buffered saline solution, sodium azide (0.02%), Tween-20 (0.1%), and DAPI (1:2500) at 4 °C, on days 3, 4, 5, 7, 10, 12, 14, 16, 18, 21 and 24 after seeding. For necrotic core measurements, 12 h prior to harvesting 1 μmol total/well DRAQ7$^{TM}$ dye (Abcam, Cambridge, United Kingdom. ab109202) was added to each well[54,69]. Fixed spheroids were set in place using low melting 2% agarose and optically cleared in 500 μL total/well high refractive index mounting solution (Quadrol 9% wt/wt, Urea 22% wt/wt, Sucrose 44% wt/wt, Triton X-100 0.1% wt/wt, water) for 2 days in a 24-well glass bottom plate (Cellvis, P24-1.5H-N) before imaging to ensure accurate measurements[70,71]. Images were then captured using an Olympus FV3000 confocal microscope with the 10× objective focused on the equatorial plane of each spheroid.

*Image processing.* Images captured with the IncuCyte S3 were processed using the accompanying IncuCyte 2020C Rev1 software (spheroid analysis type, red image channel, largest red object area per well). Area masks were visually compared with IncuCyte brightfield images to confirm accuracy. Area was converted to an equivalent radius ($r^2 = A/\pi$). Confocal microscopy images were converted to TIFF files in ImageJ and then processed with custom MATLAB scripts that use standard MATLAB image processing toolbox functions. These scripts are freely available on Zenodo with https://doi.org/10.5281/zenodo.5121093[72].

*Statistics and reproducibility.* Details of practical parameter identifiability analysis are presented in Methods: Practical parameter identifiability analysis. Data points in Figs. 2a–c and 3a, b represent measurements of individual spheroids. Semi-transparent data points in Figs. 2d–f, 3c, d, 4f represent measurements of individual spheroids, while other data points represent the mean of the measurements of either the outer radius, inhibited radius, or necrotic radius at that specific time, while error bars represent the standard deviation of those measurements. Supplementary Discussion C details the

number of measurements at each time point for each cell line and experimental data analysed during the study are available on a GitHub repository (https://github.com/ryanmurphy42/4DSpheroids_Murphy2021). We note that some measurements could not be obtained primarily due to blurring of the automated imaging, spheroids not forming properly, or spheroids losing their structural integrity at very late time. Data for these spheroids was excluded. As part of the study we compare results across different experimental designs and determine when our sample size is sufficient. Time series data and comparison of results across multiple experimental designs, presented in the manuscript and supplementary discussion, demonstrate the reproducibility of experimental findings. Randomisation and blinding was not possible.

**Reporting summary**. Further information on research design is available in the Nature Research Reporting Summary linked to this article.

## Data availability
The data sets generated during and analysed during the current study are available on a GitHub repository (https://github.com/ryanmurphy42/4DSpheroids_Murphy2021) and are summarised in the electronic supplementary information. In addition, Supplementary Data 1 contains the data presented in the figures of the main manuscript, focusing on the WM793b cell line.

## Code availability
Key computer code and all experimental data used to generate computational results are available on a GitHub repository (https://github.com/ryanmurphy42/4DSpheroids_Murphy2021). The computer code for the mathematical modelling and statistical identifiability analysis was written in MATLAB R2021b (v9.11) with the Image Processing Toolbox (v11.4), Optimization Toolbox (v9.2), Global Optimization Toolbox (v4.6), and the Statistics and Machine Learning Toolbox (v12.2).

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

## Acknowledgements

M.J.S. and N.K.H. are supported by the Australian Research Council (DP200100177). We thank Dr. Pascal Buenzli and Dr. Patrick Thomas for helpful discussions, John Blake for guidance using IncuCyte, and Dr. David Warne for guidance using the high-performance computing resource at QUT. This research was carried out at the Translational Research Institute (TRI), Woolloongabba, QLD. TRI is supported by a grant from the Australian Government. We thank the staff in the microscopy core facility at TRI for their outstanding technical support. We thank Prof. Atsushi Miyawaki, RIKEN, Wako-city, Japan, for providing the FUCCI constructs, Prof. Meenhard Herlyn, The Wistar Institute, Philadelphia, PA, for providing all of the cell lines. We thank Professor Rachel Bearon and the anonymous reviewer for their helpful comments.

## Author contributions

All authors conceived and designed the study. R.J.M. performed the research and drafted the article. R.J.M., A.P.B., and G.G. performed experimental work. All authors provided comments and approved the final version of the manuscript. N.K.H. and M.J.S. contributed equally.

## Competing interests

The authors declare no competing interests.
