## [Peer Review File · Communications Biology]

Reviewers' comments:

Reviewer #1 (Remarks to the Author):

This manuscript considers the experimental design of tumour spheroid experiments, and identifies which design enables model parameters to be confidently identified. Specifically, 3 experimental designs are considered:

- 1- Measurement of outer radius alone
- 2- Measurement of both outer radius and necrotic core
- 3- Measurement of outer, necrotic & inhibited core (the latter being identified through modification of the cell cycle in FUCCI transduced cell lines)

In all cases, the data is fit to the Greenspan mathematical model for avascular tumour spheroid growth using a maximum likelihood approach. By using a profile likelihood analysis, it is determined that the parameters found in the Greenspan model can only all be identified when the most complex experimental design (design '3') is used. As the Greenspan model includes the three regions: necrotic, inhibited & proliferative, it is perhaps not entirely surprising that parameters can only be identified when experimental measurements are made of all three regions. However, the paper explains the process of practical parameter identifiability well, and presents the first quantitative connection of the Greenspan model to data. I therefore think the manuscript is suitable for publication.

Some minor comments for the authors to consider:

The discussion about temporal resolution of the experiment was clear for experimental design 1, but needs further clarification for experimental designs 2 & 3 in the main text. For example, there is a statement that 'Additional results ... clearly show that increasing the temporal resolution may result in a worse prediction from the MLE for the time evolution of the internal structure' which is counterintuitive (& therefore interesting!), but I couldn't easily find further explanation of this within the main text.

The choice of temporal resolution ('A', 'B' or 'C') should be made explicit in fig 3.

Rachel Bearon

Reviewer #2 (Remarks to the Author):

This paper considers a simple (but seminal) PDE model, together with parameter fitting approaches, to provide rationale for optimal design criteria for tumour spheroid experiments.

The paper and the results are interesting, and the techniques and approach employed are well used. However, I have some reservations about the strength of the conclusions that the authors draw.

The premise is that by using a mathematical framework, experimental designs that provide reliable biological insight are isolated; in particular, it is concluded that cell cycle data are essential for studies that are interested in tumour spheroid structure. My reservations here are:

- (i) How does the approach guarantee "biological insight"? In large part it seems that the study (reasonably) is able to conclude which of the 3 experiments are able to give most accurate/reliable parameter estimates for the Greenspan model. This seems a rather different thing to me.
- (ii) The study limits itself to 3 specific approaches, and then concludes that one of these is "essential" to investigate spheroid structure. Surely this omits other possible approaches and this conclusion should be suitably caveated?
- (iii) The specific utility or added value of using the Greenspan model (or other model) in support of (e.g.) conclusion (ii) or the wider question of "biological insight" in (i) is not clear.

Significantly more discussion, clarification or support on these points is needed, in my view.

Minor comments:

1. I don't like the phrase 4D spheroid experiments! (Though the authors do qualify fairly precisely what they mean, it seems over-blown to me!)
2. Page 2, suggest "Greenspan mathematical model for avascular tumour growth [21] ..."
3. Page 2, typo: tranduced.
4. Ironically, the section title "Cell cycle data are informative" doesn't seem informative!
5. General comment regarding the Supplement: I think it's better if this is somewhat more self-contained. For example
 - a) it would be better to reproduce or re-draw the relevant part of Fig 1 b) S.6 the meaning of the thresholds is not clear. More detailed explanation here would help the reader.
6. Eqs S3, S4, suggest "convenient" rather than "useful".
7. The precise details of how phases (i)-(iii) are implemented (algorithmically) are not clear to me.
8. Supplement, page 7. I don't know what is intended by "We consider phase (iii) since calculations used for statistical identifiability analysis ..."
9. Formatting of S.24 makes for hard reading

Point-by-point response to reviewers for COMMSBIO-21-2500-T: Manuscript Revision

Reviewer 1 (Professor Bearon)

This manuscript considers the experimental design of tumour spheroid experiments, and identifies which design enables model parameters to be confidently identified. Specifically, 3 experimental designs are considered: 1- Measurement of outer radius alone 2- Measurement of both outer radius and necrotic core 3- Measurement of outer, necrotic & inhibited core (the latter being identified through modification of the cell cycle in FUCCI transduced cell lines)

In all cases, the data is fit to the Greenspan mathematical model for avascular tumour spheroid growth using a maximum likelihood approach. By using a profile likelihood analysis, it is determined that the parameters found in the Greenspan model can only all be identified when the most complex experimental design (design '3') is used. As the Greenspan model includes the three regions: necrotic, inhibited & proliferative, it is perhaps not entirely surprising that parameters can only be identified when experimental measurements are made of all three regions. However, the paper explains the process of practical parameter identifiability well, and presents the first quantitative connection of the Greenspan model to data. I therefore think the manuscript is suitable for publication.

Response: We thank Professor Bearon (Reviewer 1) for her positive comments and recommendation that the manuscript is suitable for publication.

Some minor comments for the authors to consider:

- (R1.1) The discussion about temporal resolution of the experiment was clear for experimental design 1, but needs further clarification for experimental designs 2 & 3 in the main text. For example, there is a statement that 'Additional results . . . clearly show that increasing the temporal resolution may result in a worse prediction from the MLE for the time evolution of the internal structure' which is counterintuitive (& therefore interesting!), but I couldn't easily find further explanation of this within the main text.

Response: We thank the Professor Bearon for highlighting this point. We now make explicit in the revised text that temporal resolutions A, B, and C, provide very similar results for Design 2 and 3 (Page 9 Lines 161–162, Page 10 Lines 191–192). Therefore, we only present temporal resolution A in the main manuscript for Design 2 and 3. We also make clear that we present the corresponding results for temporal resolutions B and C in the Supplementary Material D.2 (Page 9 Lines 161–162, Page 10 Line 190 and Lines 191–192).

We note that the statement '*Additional results . . . clearly show that increasing the temporal resolution may result in a worse prediction from the MLE for the time evolution of the internal structure*' refers to results obtained using Design 1. Specifically, Design 1 with different initial spheroid sizes and Design with the WM983b cell line (Page 7 Line 140). We now make this explicit and update the reference to the supplementary material F.1. These results do not mean that the mathematical model is incorrect. Our interpretation of these result is that the experimental data are insufficient to correctly identify the parameters in the mathematical model (Page 8 Lines 141–143), as also demonstrated in Figure 2 which shows wide 95% approximate confidence intervals for parameter such as γ .

- (R1.2) The choice of temporal resolution ('A', 'B' or 'C') should be made explicit in fig 3.

Response: We now make explicit that we show results for temporal resolution A in caption of Figure 3 (Page 11). As mentioned R.1.1 we also state in the text that results using Designs 2 and 3 are very similar using temporal resolutions A, B, and C.

Reviewer 2

This paper considers a simple (but seminal) PDE model, together with parameter fitting approaches, to provide rationale for optimal design criteria for tumour spheroid experiments.

The paper and the results are interesting, and the techniques and approach employed are well used. However, I have some reservations about the strength of the conclusions that the authors draw.

The premise is that by using a mathematical framework, experimental designs that provide reliable biological insight are isolated; in particular, it is concluded that cell cycle data are essential for studies that are interested in tumour spheroid structure. My reservations here are: (i) How does the approach guarantee “biological insight”? In large part it seems that the study (reasonably) is able to conclude which of the 3 experiments are able to give most accurate/reliable parameter estimates for the Greenspan model. This seems a rather different thing to me. (ii) The study limits itself to 3 specific approaches, and then concludes that one of these is “essential” to investigate spheroid structure. Surely this omits other possible approaches and this conclusion should be suitably caveated? (iii) The specific utility or added value of using the Greenspan model (or other model) in support of (e.g.) conclusion (ii) or the wider question of “biological insight” in (i) is not clear.

Significantly more discussion, clarification or support on these points is needed, in my view.

Response: We are glad that Reviewer 2 finds our paper and results interesting and that the techniques and approach employed are well used. We appreciate the opportunity to address their comments, and now address each point raised in turn.

(R2.1.(i)) How does the approach guarantee “biological insight”? In large part it seems that the study (reasonably) is able to conclude which of the 3 experiments are able to give most accurate/reliable parameter estimates for the Greenspan model. This seems a rather different thing to me.

Response:

We agree that our study is able to conclude which of the main 3 experimental designs, with respect to the type of measurements taken, are able to give the most accurate/reliable parameter estimates for the Greenspan model. We then point out that certain experimental designs lead to reliable biological insight, as oppose to guarantee biological insight. One important benefit of using Greenspan’s model is that all parameters in the model are physically interpretable and biologically relevant (Page 2 Lines 43–46), for example the outer radius when the necrotic region forms, R_c . As our approach identifies when we can obtain reliable estimates with low certainty of the biologically relevant parameters of Greenspan’s model, we are able to quantify the time evolution of the structure of growing spheroids while gaining insights to the contribution of underlying biological mechanisms (Page 14 Lines 229–232).

Our results also provide further biological insights, for example, experimental biologists now know that, where spheroid structure is important for the question being asked, some measurements using Design 3 is more valuable than many measurements using Designs 1 or 2. In addition, they now know that the initial number of cells used to form spheroids and temporal resolution are less important experimental design choices (Page 14 Lines 244–252).

(R2.1.(ii)) The study limits itself to 3 specific approaches, and then concludes that one of these is

“essential” to investigate spheroid structure. Surely this omits other possible approaches and this conclusion should be suitably caveated?

Response:

In this study we consider multiple experimental designs. The three that reviewer refers to are Design 1, 2, and 3 which vary by the types of measurements used. We also consider experimental designs using different temporal resolutions and different numbers of cells used to form spheroids. Based on our results one conclusion is that Design 3 is essential to investigate spheroid structure. We now caveat these statements by placing them in the context of standard experimental protocols. It is common in spheroid experiments to measure the outer radius, it is less common to measure the necrotic core and not standard protocol to measure the cell cycle. However, we reveal that cell cycle measurements are essential for accurately estimating model parameters. We agree that there may be alternative experimental designs that can be used to extract useful insights. However, we believe these approaches likely require more experimental effort and expense than the approach we use here of extending a typical experimental protocol to include necrotic core and cell cycle measurements. We note that should these alternative experimental designs be developed our framework will be relevant to their assessment (Page 14 Lines 233–243).

(R2.1.(iii)) The specific utility or added value of using the Greenspan model (or other model) in support of (e.g.) conclusion (ii) or the wider question of “biological insight” in (i) is not clear.

Response:

The Greenspan model is a seminal model in the literature. We believe the fact that it had not previously been experimentally validated is worth testing in its own right, as reviewer 1 points out. Furthermore, there are great advantages in using Greenspan’s model. Specifically, Greenspan’s model is a relatively simple mathematical model with few parameters but all with physical interpretations, as opposed to more complicated models which may have more parameters that may not be able to be determined and also may not have physical interpretations. In addition, without using a mathematical modelling and uncertainty quantification framework such that we employ here, it is impossible to quantify the value of different experiment designs, and therefore impossible to interpret and determine uncertainty of biologically relevant parameters (Page 2 Lines 43–48, Page 9 Lines 156–158, Page 14 Lines 237–239).

Finally, the work presented here lays the foundation for future work which could incorporate drug therapies and therefore be able to quantify the effectiveness of cancer treatments, possibly across different experimental designs (Page 15 Lines 273–274). This another advantage and contributing factor to choosing the Greenspan model, as opposed to a model using only the outer radius and/or necrotic core measurements, since many drugs are effective at different phases of the cell cycle so understanding the time evolution of the outer radius, inhibited radius, and necrotic radius will be of great value (Page 15 Lines 277–279).

Minor comments:

(R2.2) I don’t like the phrase 4D spheroid experiments! (Though the authors do qualify fairly precisely what they mean, it seems over-blown to me!)

Response: We thank the reviewer for stating that we qualify fairly precisely what we mean with the phrase 4D spheroid experiments but we respectfully disagree and believe

that our terminology is not over-blown. Our experimental measurements allow us to identify the position (x, y, z) and cell cycle status, which can be thought of as a measure of the time since entering the cell cycle for a freely-cycling cell, giving rise to the term 4D tumour spheroid experiments (Page 3 Lines 62–63). Note that traditional spheroid experiments without cell cycle labelling could use nuclear staining to identify position but not cell cycle status. This means that our experiments provide more information than a traditional 3D spheroid experiment. We think the simplest way to make this clear is to refer to the experiments as “4D tumour spheroid experiments”, and we note that Referee 1 did not object to this terminology. Therefore, unless the editor explicitly prefers that we drop this terminology, we prefer to maintain it because it succinctly describes the nature of our experiments.

(R2.3) Page 2, suggest “Greenspan mathematical model for avascular tumour growth [21] ...”

Response: We agree and update the text accordingly (Page 2 Line 34).

(R2.4) Page 2, typo: tranduced.

Response: We now correct this typographical error (Page 2 Line 53).

(R2.5) Ironically, the section title “Cell cycle data are informative” doesn’t seem informative!

Response: We acknowledge that this section title should be more descriptive. We now update this section title to “*Cell cycle and necrotic core measurements reveal time evolution of internal spheroid structure*” (Page 9 Lines 147–148). We also update the corresponding section titles in the supplementary material (Supplementary Page 45 Lines 351–352, Supplementary Page 51 Lines 393–394, Supplementary Page 51 Lines 446–447)).

(R2.6) General comment regarding the Supplement: I think it’s better if this is somewhat more self-contained. For example a) it would be better to reproduce or re-draw the relevant part of Fig 1 b) S.6 the meaning of the thresholds is not clear. More detailed explanation here would help the reader.

Response: We now make the supplementary material more self-contained.

For point (a), we add an additional figure to the model description (Supplementary Figure S1 on Supplementary Page 3) that presents an example realisation of Greenspan’s model, the key mechanisms governing Greenspan’s model, and how the nutrient thresholds are defined. For additional results in the supplementary material we do not redraw parts of Figure 1 since this would introduce many additional figures and the supplementary results sections are labelled and described to allow direct comparison with the results presented in the main manuscript.

For point (b) the nutrient threshold for inhibition, c_i , and nutrient threshold for necrosis, c_n , implicitly define the inhibited radius, $R_i(t)$, and necrotic radius, $R_n(t)$ in Greenspan’s model. We update the text to make this clearer and also now make explicit reference to Figure 1g for a schematic representing these relationships and the corresponding panel of the new Supplementary Figure S1c (Supplementary Page 4 Lines 61–64).

(R2.7) Eqs S3, S4, suggest “convenient” rather than “useful”.

Response: We now update the text to state convenient rather than useful (Supplementary Page 4 Line 57).

(R2.8) The precise details of how phases (i)-(iii) are implemented (algorithmically) are not clear to me.

Response: We agree that in Supplementary Material Sections A and B we outline in some detail the process to solve the governing equations and perform statistical identifiability analysis but do not provide the precise algorithm. We now highlight that code used to produce the results in this work is freely available on GitHub, where precise details of the implementation can be viewed (Page 20 Lines 398–402, Supplementary Page 8 Lines 138–139, Supplementary Page 13 Lines 272–273).

(R2.9) Supplement, page 7. I don't know what is intended by "We consider phase (iii) since calculations used for statistical identifiability analysis ..."

Response: We agree that this was confusing. Accordingly, we now update this text to "*While we do not experimentally observe spheroids forming in phase (iii), here we include how to initialise Greenspan's model in phase (iii) for completeness and since calculations used for statistical identifiability analysis (Supplementary Material B.1) may evaluate the likelihood of starting in phase (iii).*" (Supplementary Page 8 Lines 145–148).

(R2.10) Formatting of S.24 makes for hard reading

Response: To improve the formatting for Equation S.24 we now remove the text under each term in the equation and rewrite this in the text directly below the equation (Supplementary Page 9 Lines 188–189).

REVIEWERS' COMMENTS:

Reviewer #1 (Remarks to the Author):

I am satisfied all previous comments have been addressed & so recommend the paper as now suitable for publication.

Reviewer #2 (Remarks to the Author):

The authors have addressed all my comments, and I am happy to recommend publication with no further revision.